# The Microbial Diversity and Biofilm-Forming Characteristic of Two Traditional Tibetan Kefir Grains

**DOI:** 10.3390/foods11010012

**Published:** 2021-12-21

**Authors:** Xiaomeng Wang, Wenpei Li, Mengjia Xu, Juanjuan Tian, Wei Li

**Affiliations:** College of Food Science and Technology, Nanjing Agricultural University, Nanjing 210095, China; 2019208028@njau.edu.cn (X.W.); 9181810511@njau.edu.cn (W.L.); t2020170@njau.edu.cn (M.X.); 2018208012@njau.edu.cn (J.T.)

**Keywords:** kefir grains, high-throughput sequencing, extracellular polysaccharide (EPS), biofilm

## Abstract

In this study, a high-throughput sequencing technique was used to analyze bacterial and fungal diversity of two traditional Tibetan kefir grains from Linzhi (K1) and Naqu (K2) regions. Comparative bioinformatic analyses indicated that *Lactobacillus kefiranofaciens*, *L. kefiri* and *Kluyveromyces marxianus* were the main dominant strains in K1 and K2. In order to research the relationship of the growth of kefir grains, the biofilm and the extracellular polysaccharides (EPS) produced by microorganisms, the proliferation rate of kefir grains, the yield and chemical structure of EPS and the optimal days for biofilm formation were determined. The results showed that the growth rate, the yield of EPS and the biofilm formation ability of K1 were higher than K2, and the optimal day of their biofilm formation was the same in 10th day. Additionally, the live cells, dead cells and EPS in biofilm formation of K1 and K2 were observed by fluorescence microscope to clarify the formation process of kefir grains. To determine the influence of microbial interactions on biofilm and the formation of kefir grains, the essential role of microbial quorum sensing needs further attention.

## 1. Introduction

Kefir is an acidic, viscous, somewhat effervescent and slightly alcoholic traditional dairy beverage which is obtained by the fermentation of milk with kefir grains [1,2,3]. Kefir grain is elastic, slimy, and generally ranges from 1 to 3 cm in size with an appearance like cauliflower or popcorn. Kefir grains are a fascinating biological entity which are made up of a natural matrix of extracellular polysaccharide (EPS) and complex microbiota including lactic acid bacteria (LAB), yeast and acetic acid bacteria (AAB) [4,5,6]. Many researchers have evaluated kefir grains with different microbial diversity from different countries including Argentine [7], Brazilian [8], Irish [9], Turkish [5] and Tibetan grains [10]. A variety of microorganisms were found in different kefir grains, and the main genus was *Lactobacillus*, while low abundance genera were also found in grains from different sources, such as *Leuconostoc*, *Lactococcus* and *Streptococcus*. A composition of yeasts (*Kluyveromyces*, *Saccharomyces* and nonpathogenic strains of *Candida*) and some AAB were also revealed in different kefir grains [11]. In fact, the total number of microorganisms isolated from various kefirs is estimated at over 300 species emphasized their complex and diverse composition [7,12]. Although in distinct regions of the country, the microbial diversity of kefir grains was still different. Zhou et al. have conducted research on Tibetan kefirs and found *Leuconostoc mesenteroides*, *L. lactis*, *L. kefiri*, *L. casei*, *K. marxianus*, *S. unisporus*, *S. cerevisiae* and *Candida humilis* in Tibetan kefirs [13]. Gao and Zhang [10] isolated *L. kefiranofaciens*, *S. thermophilus*, *L. kefiri* and *L. paracasei* from Tibetan kefirs which exhibited the intraspecies genotypic diversity. Linzhi and Naqu are two typical regions in Tibet. Different microbial communities of kefir grains are likely to establish in Linzhi and Naqu due to the different climate and environment.

Tibetan kefir grain possessed health benefits including a cholesterol-lowering capacity [14], antioxidant capacity [15] and colonisation and gut flora modulation capacity [16]. They are initially small but increase in size during fermentation, and the grains can only grow from pre-existing grains. Therefore, understanding the formation process of kefir grains acts is critical in its artificial production and industrial application. Kefir grains increase their weight as a consequence of the growth of microorganisms and the biosynthesis of grain components. The synthesis of EPS is also necessary to increase the biomass of kefir grains. Many strains of LAB from kefir grains can produce EPS, which can be used as capsular polysaccharides (c-EPS) to attach to the surface of bacteria or released into the surrounding medium as released polysaccharides (r-EPS) [17]. c-EPS attached to the cell wall of LAB is generally a heteropolysaccharide with large molecular weight and insoluble characteristic. When the c-EPS was extracted by different methods, the bond type was destroyed accordingly, and the structure of a few EPS released to the medium was consistent with the r-EPS produced by LAB [18]. The microorganism of kefir and the EPS produced by LAB associated with the grains could be considered as a biofilm [19]. Bacterial biofilm is structured communities of cells which encapsulated in an extracellular polymeric substance comprised of EPS, proteins and DNA. Biofilm formation by LAB may assist cells to resist environmental stress such as higher levels of acetic acid and ethanol [20]. In recent, some studies have also shown that biofilms can promote the formation of kefir grains [21]. Dong et al. [22] emphasized that the establishment of biofilm contributed the main role in the formation processes of kefir grains. However, at present, there is still a lack of knowledge concerning the role of EPS-producing strains of LAB and their biofilm in the formation of Tibetan kefir grains.

This study aims to elucidate the microbiological composition about kefir grains and milks collected from two traditional regions of Linzhi and Naqu of Tibet. In addition, bacteria and yeast from kefir grains were isolated and identified. Furthermore, to research the relationship of the growth of kefir grains, the biofilm and EPS produced by microorganisms, the proliferation rate of kefir grains, the structural characteristics of kefir EPS and the process of kefir biofilm generation were also evaluated in this study. Investigating the formation process of kefir grains has a major impact on the artificial expansion and industrial application of kefir grains.

## 2. Materials and Methods

### 2.1. Activation of Tibetan Kefir Grains

Kefir grains were collected from two traditional regions of Tibet in Linzhi (K1) and Naqu (K2), China. The grains were activated at 28 °C for 24 h in the sterile whole milk three times. About 5% (*w*/*v*) of activated grains of each sample were inoculated into 500 mL sterile whole milk, and the produced kefir milks were obtained immediately after fermenting at 28 °C for 24 h.

### 2.2. Macroscopic Observation and Microscopic Observation of Kefir Grains

The two activated kefir grains were rinsed with sterilized water, and then their macroscopic structure and microscopic composition were observed. The microbial observation of the kefir grains was observed under the microscope by the negative staining method. The conventional smears were used to stain kefir microorganism with 0.5% (*m*/*v*) crystal violet staining solution for 2 min after they were air-dried, then 20% (*w*/*v*) CuSO_4_ solution was used to elute to observe.

### 2.3. High-Throughput Analysis of Microorganism Diversity on Kefir Grains and Kefir Milks

#### 2.3.1. Bacterial and Fungal Total DNA Extraction and Polymerase Chain Reaction (PCR) Amplifycation

The microbial DNA was extracted directly from the two kefir grains (K1-G and K2-G) and kefir milks (K1-M and K2-M) using a Power Food^TM^ Microbial DNA Isolation Kit (Mo Bio Laboratories, Carlsbad, CA, USA). Specifically, 1 g of each kefir grain was ground, while 1.5 mL of each kefir milk was centrifuged at 12,000 g for 3 min. Then each sample was processed according to the instructions. The V3-V4 regions of 16S rDNA genes were amplified using universal primers 338F (5′-ACTCCTACGGGAGGCAGCAG-3′) and 806R (5′-GGACTACHVGGGTWTCTAAT-3′). The ITS regions were amplified using the primers ITS1F (5′-CTTGGTCATTTAGAGGAAGTAA-3′) and ITS2R (5′-GCTGCGTTCTTCATCGATGC-3′).

#### 2.3.2. High-Throughput Sequencing and Bioinformatic Analysis

16S rRNA and internal transcribed spacer (ITS) genes were performed on Illumina Miseq platform at Majorbio Inc. (Shanghai, China). In order to obtain the stability data, the original data extraction unit and operational classification unit (OTU) cluster analysis were performed in each group. Shannon index, Chao index, Simpson index, Venn analysis, principal coordinate analysis and principal coordinate analysis (PCoA) score plots were conducted via the R package for the comparison of K1 and K2.

### 2.4. Isolation of Bacteria and Yeast from Kefir Grains

The method of sub-culturing kefir grains in milk as mentioned in the Section 2.1. The medium of deMan Rogosa Sharpe (MRS) broth and potato dextrose agar (PDA) broth was allowed for the accumulation of LAB and yeasts. Briefly, 1 g of grain was ground in a sterile ceramic mortar, further 10-fold dilutions of suspension of kefir grains and kefir milks were prepared. Then, 0.1 mL of each 10^−1^–10^−7^ dilutions were plated (in duplicate) with a spreading rod onto petri dishes with agar medium incubated at 28 °C. Repeat the process of streaking culture of a single colony until the colony as a pure strain under microscope observation. The DNA of isolated strains was further extracted as Section 2.1 mentioned. A fragment 16S rDNA of bacteria was amplified by forwarding primer 27F: 5′-AGAGTTTGATCCTGGCTCAG-3′, and reverse primer 1492R: 5’-GGCTACCTTGTTACGACTT-3’. A fragment 16S rDNA of yeast was amplified by forwarding primer NL1: 5’-GCATATCAATAAGCGGAGGAAAAG-3’, and reverse primer NL4: 5’-GGTCCGTGTTTCAAGACGG-3’. The PCR products were purified and then submitted to sequence.

### 2.5. Determination of the Proliferation Rate of Kefir Grains

The cleaned kefir grains were inoculated into milk at 28 °C for 48 h, and then they were filtered with a colander and washed with sterile water for bacterial passage. Finally, the grains were weighed. This method was repeated for another passage to calculate the proliferation rate of kefir grains.
(1)Proliferation rate (%)=(Mn−M0) / M0×100   

*Mn*: The weight of kefir grains after the nth passage.

*M*0: The weight of the initial inoculated kefir grains.

### 2.6. The Yield and Structural Investigation of Kefir EPS

The r-EPS of K1 and K2 was extracted according to our previous method with minor modifications [23]. Briefly, fermentation milk was separated by centrifugation at 12,000 rpm for 15 min after culturing at 28 °C for 48 h. Then 80% (*w*/*v*) trichloroacetic acid (TCA) was added to the final concentration of 4% (*w*/*v*). The supernatant was kept at 4 °C for 6 h, and the precipitation was discarded by centrifugation (12,000 rpm for 10 min). The supernatant was mixed with three volumes of absolute ethanol overnight at 4 °C. After centrifugation, the crude r-EPS precipitation was completely dissolved in a distilled water and dialyzed using dialysis bag (molecular weight cut-off 8000–14,000 Da, Solarbio Co., Ltd., Beijing, China) for 72 h at 4 °C. Finally, the supernatant was lyophilized to obtain r-EPS. The c-EPS was extracted by ultrasonic assisted extraction method [18]. In order to obtain the microorganism, the MRS medium was used to activate kefir grains. Firstly, the fermentation was centrifuged at 8000 rpm for 10 min, and the precipitation was washed twice with 0.85% (*w*/*v*) NaCl, then the precipitation was resuspended in 1 M NaCl and sonicated at 40 W, 4 °C for 3 min. The following steps were the same as the extraction of r-EPS.

#### 2.6.1. Monosaccharide Composition Analysis of Kefir EPS

The monosaccharide composition of EPS of K1 and K2 was determined as our previous method with minor modifications [24]. 5 mg of EPS was hydrolyzed with 2 mL of 2 M trifluoroacetic acid (TFA) at 120 °C for 2 h. The residual TFA was removed through rotary evaporation. Subsequently, the residue was dissolved in deionized water and mixed with 0.5 M 3-methyl-1-phenyl-2-py-razolin-5-one (PMP) solution and 0.3 M NaOH solution, and then reacted at 70 °C for 30 min. After neutralization with 0.3 M HCl, the solution was added to 2 mL of deionized water and 4 mL of chloroform followed by the vortex. The upper aqueous phase was analyzed by Waters 2695 HPLC system equipped with a photo-diode array (PDA) detector and a C_18_ column (250 mm × 0.4 mm i.d., 0.5 μm) at a flow rate of 0.8 mL/min at 25 °C. The mobile phase consisted of 0.1 M ammonium acetate, acetonitrile and tetrahydrofuran at the volume ratios of 81:17:2.

#### 2.6.2. Fourier-Transform Infrared (FT-IR) Analysis of Kefir EPS

The FT-IR spectra of EPS of K1 and K2 were recorded on a fourier-transform infrared spectrophotometer (Bruker Tensor-27, Bruker Co., Ettlingen, Germany) with the frequency range of 4000–500 cm^−1^ for the determination of the functional groups.

### 2.7. Determination of the Optimal Days for Biofilm Formation

The cleaned kefir grains were inoculated into milk at 25 °C for 48 h, and then 200 µL fermented milk were resuspended into 0.85% (*w*/*v*) NaCl solution to obtain precipitations. The precipitations were resuspended in sterile MRS, and then 1 mL of resuspension was added to the 48-well plate with the incubation at 28 °C for different times. After that, the plate was washed by 0.85% (*w*/*v*) NaCl solution. It was dried at 42 °C for 30 min and stained with 1 mL of 0.1% (*v*/*v*) crystal violet solution for 5 min. Then the biofilm on the wells was dissolved with 1 mL of 95% (*v*/*v*) ethanol, and the OD_560_ value was measured to indicate the biofilm formation ability of kefir grains. Similarly, 1 mL of resuspension was inoculated in a 48-well plate for 2, 4, 6, 8, 10 and 12 days to determinate the optimal days of biofilm formation.

### 2.8. Fluorescence Microscope Observation in the Formation of Kefir Biofilm

Live cells, dead cells and EPS of kefir biofilms were inoculated in 48-well plates for different days (4, 6, 8 and 10) and observed by fluorescence microscope. The numbers of live and dead cells on the biofilm were fluorescently stained with Syto9/PI reagent, and the yield of EPS of microorganisms was fluorescently stained with FITC-ConA/PI reagent. Briefly, a mixture of 2.5 µM Syto9 and PI solution was used to dye bacterial cells for 30 min in the dark, and the excitation wavelengths of Syto9 and PI were 488 and 561 nm. Finally, live and dead cells will be imaged as green and red, respectively. For FITC-ConA/PI labeling method: the biofilm was fixed with 2.5% (*v*/*v*) glutaraldehyde for 1 h, so that all the strains in the membrane can be died and stained with PI dye. A mixture of FITC-ConA and PI dye was used to dye the membrane for 30 min in the dark. The excitation wavelengths of FITC-ConA and PI were 488 and 561 nm, and then EPS and dead bacteria will be imaged as green and red, respectively.

### 2.9. Statistical Analysis

The obtained results were analyzed using SPSS version 16.0 (SPSS Inc., Chicago, IL, USA). All data were analyzed by one-way analysis of variance (ANOVA). Significant differences (*p* < 0.05) were determined by Tukey’s multiple range test to compare the differences among various groups.

## 3. Results and Discussion

### 3.1. Macroscopic Observation and Microscopic Observation of Kefir Grains

The macroscopic of kefir grains from Tibet was shown in Figure 1. In this study, small kefir grains of K1 and K2 were unfolded in a similar form to thick films. Overall, both kefir grains from Linzhi and Naqu regions had irregular appearances with cauliflower-like. The two kefir grains were different in size. K1 was significantly larger than K2, while the tightness of K1 was better than K2. The microscopic observation of K1 and K2 under an optical microscope was shown in Figure 1(C1,C2). Different shapes of bacteria and yeast were dyed. The bacteria (1a and 2a) and yeast (1b and 2b) appeared purple under an optical microscope, while the capsules were seen as an unstained and colorless halo surrounding the cells against a light violet-gray background.

### 3.2. Sequencing Results and Alpha Diversity Analysis

A total of 149,560 high-quality valid bacterial 16S rDNA gene sequences and 122,764 high-qualities valid fungal ITS rDNA gene sequences were obtained after the restrictive filtering process from the four analyzed samples (Table 1). At the high threshold identity cut-off level of 97% sequence similarity, a total of 52 and 45 OTUs were selected from K1 and K2, respectively. All analyses rely on OTUs detected in kefir grains. Shannon index showed an appropriate sequence of both richness and evenness of bacterial microbiota in kefir grains [25]. Additionally, Shannon index ranged from 0.11 to 1.35, which indicated that most bacteria in kefir grains were found. Shannon index in each group of K2-G and K2-M indicated the most richness of bacterial species in them. Chao index and Simpson index were supported the results described above. Totally, most fungal species were captured, and the results indicated that the sequencing depth was suitable for further analysis.

### 3.3. Microbial Community Composition Analysis

A Venn diagram of the bacteria and fungi was shown in Figure 2(A1–A4) after comparing the measured sequences and OTU analysis. For bacteria, the same number of species between K1-G and K1-M was five, and the number of identical species between K2-G and K2-M was nine. For fungi, six identical species were found to exist in both K1-G and K1-M, and the same number of species of K2-G and K2-M was eleven. The Venn diagram indicated that the diversity of bacteria and fungi of two kefir grains was quite different from that of two kefir milks.

For taxonomic analysis, *Lactobacillus* was the predominant species (98%) commonly encountered in two kefir grains. Similar results were reported by Wang et al. who reported that more than 98% of the bacterial population in kefir grains was *Lactobacillus* [26]. At species level (Figure 2(B1,B2)), *Lactobacillus kefiranofaciens*, *Bifidobacterium psychraerophilum* and *L. kefiri* were the top three abundant species (up to 99.93% of all species identified) in K1, while *L. kefiranofaciens*, *Acetobacter lovaniensis*, *Enterococcus durans* (up to 95.32%) were the top three abundant species in K2. *A. lovaniensis* and *E. durans* were only found in K2-M at a level of 33.97% and 17.94%, respectively. Our findings revealed that both *L. kefiranofaciens* and *L. kefiri* have presented in K1 and K2. *L. kefiranofaciens* was the dominant bacterial species in K1 and K2, and this species comprised almost half of the microbial species in grains. A previous study indicated that *L. kefiranofaciens* and *L. kefiri* were the key LAB species in kefir grains, and they co-aggregated with other organisms and components in the milk to form the grains [27]. Other studies also reported the dominant bacterial species of Turkish kefir grains was *L. kefiranofaciens* [28]. *B. psychraerophilum* was the second *Lactobacillus* in K1-M, which indicated that K1 cultivated much *B. psychrophilus* during the fermentation process at a relatively low temperature. *A. lovaniensis* and *E. durans* were the most abundant species in K2-M. As a special kind of bacteria in kefir grains, *Acetobacter* is common, but the proportion is lower than LAB and yeast. Kumar et al. [29] identified *A. lovaniensis* in kefir samples using metagenetic analysis targeting the 16S and 26S rDNA DNA fragments.

In addition to the large and variable bacterial population in kefir grains, an abundant yeast population that existed in a symbiotic relationship with the bacteria were detected [30]. Yeast can produce ethanol and carbon dioxide for symbiosis between the microorganisms of the kefir grains. In our study, two main fungal species of *K. marxianus* and *Kazachstania turicensis* were classified. *K. marxianus* had the advantage in K1, while *K. turicensis* was richer in K2 (Figure 2(B1,B2)). The relative abundances for *K. marxianus* ranged from 0.21% to 95.57% between K1-G and K1-M, and 7.65% to 19.44% between K2-G and K2-M, respectively. *Kluyveromyces* makes up the majority or entirety of the lactose utilizing yeast population, with *K. marxianus* and *K. lactis* being the two most common species [31]. *K. marxianus* as the main species, its existence, quantity, activity and other factors affect the unique flavor and quality of kefir to a certain extent [32]. Fadda et al. [33] isolated seven strains of *K. marxianus* from kefir grains and proved that they have probiotic properties. *K. turicensis* was the most abundant species in K2-G, while the content of it in K2-M was decreased when cultured in milk for 48 h. Wang et al. [27] reported that although the content of *M. Kazakhstan* cultured in kefir milk was less represented in Tibet in China, it had high co-aggregation ability when cultivated together with *L. kefiranofaciens* and *L. kefiri*, thus participated in the early formation of kefir grains. Additionally, two mycotic species (*Cutaneotrichosporon curvatus* and *C. cutaneum*) were found in K2-M with contents below 10%.

PCoA score plots of bacteria and fungi from all samples were shown in Figure 2(C1,C2). It showed that the species composition between K1-G and K1-M were similar, while K2-G and K2-M had similar species composition. The abscissa and ordinate explained the bacteria composition difference value of 97.73% (PC1: 54.67% and PC2: 43.06%) and the fungal composition difference value of 98.65% (PC1: 60.55% and PC2: 38.10%) respectively, which indicated that the microbial composition had changed significantly. The analysis of kefir grains and kefir milks showed that microorganisms at lower abundance in the grains could become dominant in the fermented milk with the cultivation time increasing.

### 3.4. Isolation of Bacteria and Yeast from Kefir Grains

Separation and identification results showed that three strains of two types of bacteria (*L. kefiri* and *L. kefiranofaciens*) and one yeast strain of *K. marxianus* were isolated from K1-M (Table 2). Three bacteria strains of two species (*L. kefiri* and *L. kefiranofaciens*) and one yeast strain of *K. marxianus* were identified from K1-G. Consistent with the results of our study, *L. kefiranofaciens* was also reported as the dominant bacterial species in the overseas kefir grains from Turkish [28]. In addition, Moradi and Kalanpour [34] reported that *L. kefiranofaciens* plays a significant role in the outer EPS layer of kefir grains. Additionally, five bacteria strains (*E. durans*, *Fructobacillus fructosus*, *L. kefiri*, *L. paracasei* and *A. fabarum*) and one yeast strain of *K. marxianus* were identified from K2-M, and two bacteria strains (*E. durans* and *L. kefiri*) and one yeast strain of *K. marxianus* were isolated from K2-G. The strains isolated from kefir milks were more abundant than kefir grains, which was consistent with the results of high-throughput sequencing. The presence of this difference may be that the rich nutrients in milk promote the growth of different strains. Moreover, individual screening affected their viability, and some strains only lived in a co-cultivation environment [13].

### 3.5. Determination of the Proliferation Rate of Kefir Grains

The proliferation rate of K1 and K2 were shown in Figure 3A. It could be seen that the weights of the two kefir grains increased with the increase of cultivation days, and the proliferation rate of K1 was greater than that of K2. The proliferation rates of K1 and K2 were 96.67% and 68.18%, respectively, on the 9th day of culture, indicating that the growth rate of kefir grains from Linzhi and Naqu regions of Tibet was not the same.

### 3.6. Determination of the Yield of Kefir EPS

The yield of kefir r-EPS produced by fermentation of K1 and K2 were 105 ± 10.27 and 74 ± 15.32 mg/L, respectively. The content of r-EPS produced by K1 fermentation was significantly higher than that of r-EPS produced by K2 (*p* < 0.05). In addition, both kefir grains could produce c-EPS from the observation of the microbial capsule in Figure 1C, and the yield of kefir c-EPS produced by K1 and K2 were 71 ± 4.25 and 53 ± 3.48 mg/L, respectively. It should be noted that although the results showed the extraction amounts of c-EPS was very small, the yields of c-EPS extracted by existing methods were much lower than the actual yields because of the tightly bounding of it to the surface of bacteria.

### 3.7. Monosaccharide Composition Analysis of Kefir EPS

The profile of monosaccharide composition in Figure 3B revealed that r-EPS of K1 and K2 were mainly composed of glucose, galactose, mannose and rhamnose in molar ratios of 4.7:2.7:1.6:1 and 4.0:1.9:1.1:1, respectively. In addition, monosaccharide composition of r-EPS of K1 and K2 was the same as that of c-EPS of K1 and K2 (data not shown). Ghasemlou et al. [35] reported that EPS isolated from kefir grains consisting of glucose and galactose with molar ratios of 1.0:1.1. Prado et al. [36] indicated the monosaccharide composition of the EPS via fermentation of Tibetan kefir possessed higher proportions of galactose, glucose and mannose, and very small proportions of rhamnose and arabinose. The difference for the composition of monosaccharides might be caused by the r-EPS other strains produced.

### 3.8. FT-IR Analysis of Kefir EPS

The FT-IR analysis of EPS of K1 and K2 was shown in Figure 3C, which exhibited typical EPS absorption peaks in the range of 4000–500 cm^−1^. The EPS of K1 and K2 exhibited wide and strong absorption peaks around 3400 cm^−1^ due to the stretching vibration of the hydroxyl group (O-H). The weak absorption peaks near 2933 cm^−1^ and 1437 cm^−1^ were the stretching vibrations of the C-H bond group, and a broad band located at 1000–1200 cm^−1^ assigned to overlapped C-O, C-C stretching and C-OH bending modes [37,38,39].

### 3.9. Determination of the Optimal Days for Biofilm Formation of K1 and K2

Biofilm is a consortium of microorganisms in which cells stick to each other and also to a surface. On the biofilm, cells are embedded and protected within an extracellular matrix that is composed of EPS produced by the microorganism [19,21]. Figure 1(A1,A2,B) showed the unfolded images of fallen kefir grains with curling thick films during the growth process of kefir grains. The presence of curling thick films, complex microorganisms and EPS of kefir grains indicated the formation of kefir grains was inseparable from the biofilm. Biofilms of kefir grains were formed when K1 and K2 were cultivated on 2 d, 4 d, 6 d, 8 d, 10 d and 12 d, and OD_560_ nm value was calculated in Figure 4 to detect the biofilm-forming ability. Both K1 and K2 had a strong biofilm formation ability when cultured for 4 d, 6 d, 8 d, 10 d and 12 d. The biofilm-forming ability for the two grains increased first and then decreased, and reached the maximum on the 10th day of culture. Piermaria et al. [40] found that EPS isolated from kefir grains showed a strong ability of biofilm formation at the concentration of 10 g/kg, with a transparent phenotype. Prado et al. [36] reported that biofilm constitute a protected growth mode that enables microorganisms to survive in unfavorable environments, and the protected effect may be one of the reasons for the increase of biofilm in the first 10 days. Furthermore, this is also related to the dynamics of biofilm development that EPS can be deposited directly on a surface for attachment or on microbial cell surfaces to promote the initial adhesion. Koo and Yamada [41] found that with the growth of biofilm, the EPS embeds microorganisms locally in a polymer matrix. These matrices help microorganisms to form ordered cell clusters or aggregates that can form small colonies varying in shape and size. This may be the reason that the biofilm-forming ability began to decline after 10 days. In general, the biofilm-forming ability of K1 was stronger than that of K2, which could be related to the relatively higher content of EPS obtained from K1.

### 3.10. Fluorescence Microscope Observation for Biofilm Formation of K1 and K2

Fluorescence microscope observation of all the bacteria cells and EPS on the surface of the biofilms formed by K1 and K2 was determined after 4, 6, 8 and 10 days of cultivation. It can be seen from Figure 5 that the total number of microorganisms on the biofilm increased with the increase of cultivation days. On the first 4 days of culture, the counts of all cells on the biofilm of K1 and K2 were increased, and the counts of live cells were higher than the dead cells. In addition, the numbers of dead cells on the biofilms were more than live cells on the 6th day of culture. Compared with the biofilm on the 8th day of cultivation, a decrease in the amounts of live cells of K1 and K2 was evident, and the counts of dead cells were increased significantly on the 10th day. Moreover, the similarity of K1 and K2 is the number of live cells still accounts for a small part of the cell membrane on the 10th day of cultivation. A growing number of live cells in the process of biofilm formation may be related to the increased cells which can promote adhesion of microorganisms and benefit the formation of biofilm [21]. The existence of dead cells on the biofilm may be due to the change of the environment the microorganisms lived in, and the existence of dead cells was conducive to the growth and reproduction of some microorganism. In addition, the content of extracellular substances of microorganism increased with the increase of cultivation time of biofilm. These extracellular substances would envelop some microorganisms and affected the metabolic processes of these microorganisms, thus the counts of dead cells increased. 

FITC-ConA can be combined with alginate, and alginate is the main component of EPS in biofilm. Therefore, the changes in EPS content in biofilms for different days can be labeled using FITC-ConA. After culturing the biofilm for 4, 6, 8 and 10 days, the contents of EPS on the biofilm increased with the increase of cultivation time, especially on the 6th day, a dense distribution of EPS can be seen on the biofilm (Figure 5). While on the 10th day, a part of the cells was enfolded with EPS, thus an orange biofilm was observed. Similar results were reported by Marshall et al. [42], who found that the content of EPS on the biofilm was also increased with cultivation time increasing. This is because that EPS is conducive to the adsorption of microorganisms in the matrix and the adhesion between microorganisms, forming a three-dimensional biofilm structure [21]. Microorganism and EPS play a key role in the formation of biofilms from the above results. Therefore, as the bacteria and EPS on the biofilm accumulated, the biofilm continued to grow. However, the amount of biofilm reached the maximum when the biofilm was cultured to the 10th day, and the amount of biofilm began to decrease after 10 days as concluded in Section 3.9. Among the culturing process of kefir grains, the formation of biofilms was ultimately affected the proportion of live and dead microorganisms and the EPS. It also explained that microorganisms and EPS use biofilm as a medium to act a significant role in the formation of kefir grains.

## 4. Conclusions

In conclusion, the microbial community composition of K1 and K2 was significantly different by high-throughput sequencing technology. We also explored the role of biofilm in the formation of kefir grains. Results showed that the growth rate, the yield of EPS and the biofilm formation ability of K1 were higher than K2, and the optimal day of their biofilm formation was the same in 10th day. Additionally, all cells and EPS on the surface of biofilm were observed, and the amount of biofilm with all microorganisms and EPS of K1 and K2 reached the maximum when the biofilm was cultured to the 10th day. The results showed that microorganisms, EPS produced by LAB and the biofilms have a direct impact on the formation of kefir grains. This study confirmed the importance of microorganisms and EPS for the production of biofilms and further paved the way for the formation of kefir grains. Moreover, it can provide a basis for the expansion cultivation and industrial application of kefir grains. In the future, we will focus on the bacteria combination which influencing these parameters to create a new formulation of kefir grains with desirable characteristics.

## Figures and Tables

**Figure 1 foods-11-00012-f001:**
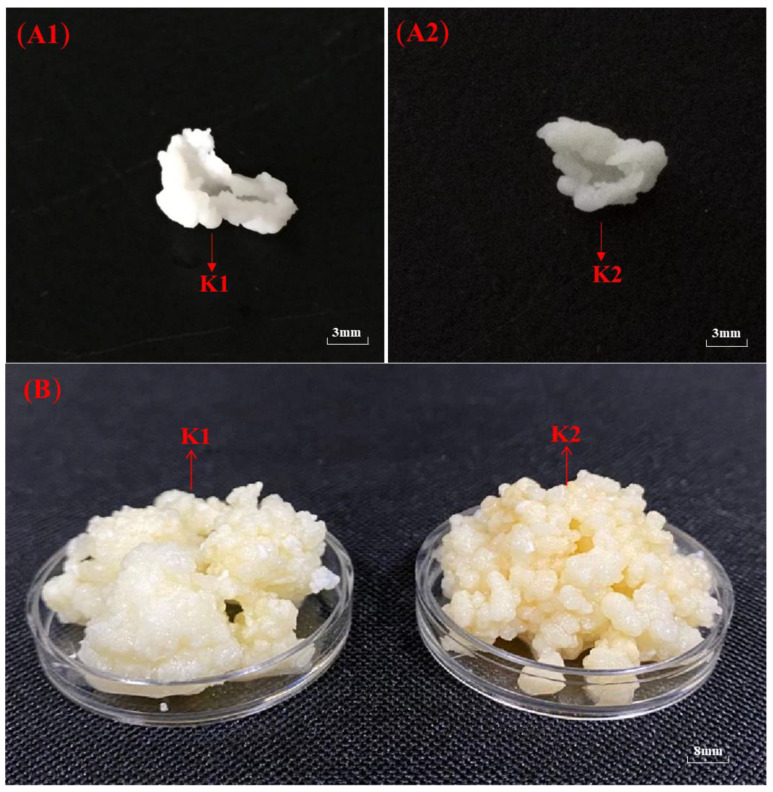
The integrated (**A1**,**A2**) and expanded partial (**B**) macroscopic observation and microscopic observation (**C1**,**C2**) of K1 and K2 from Tibet.

**Figure 2 foods-11-00012-f002:**
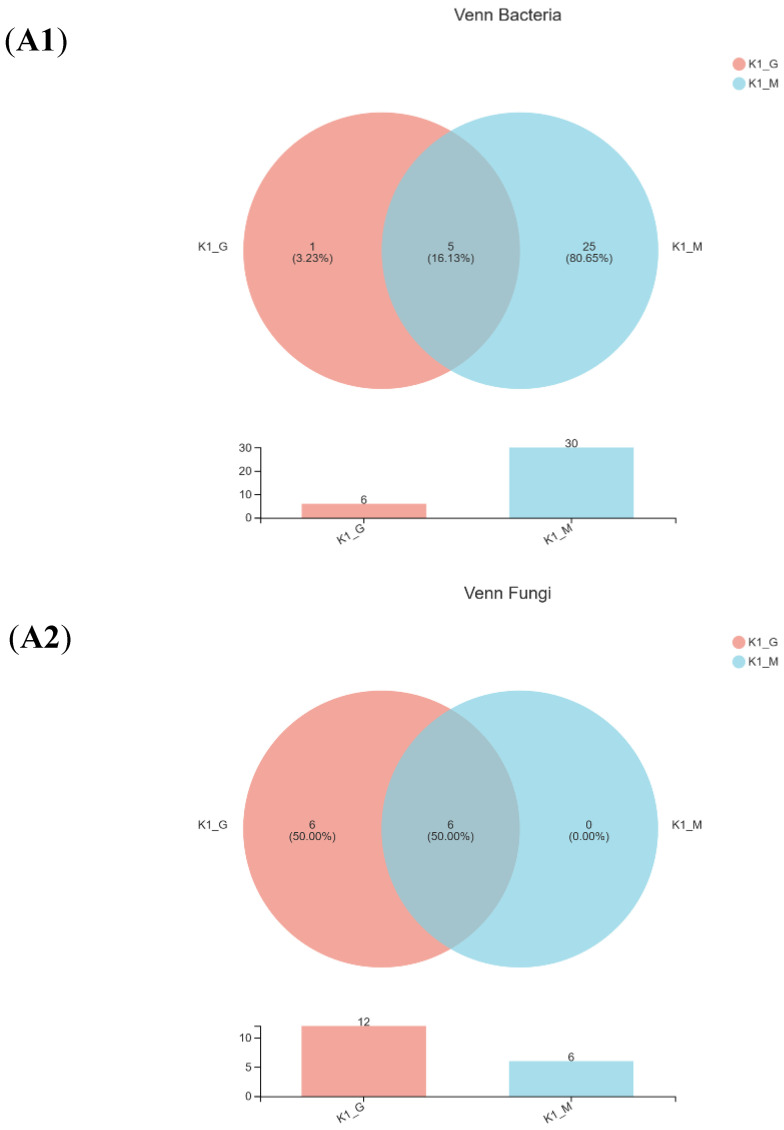
Venn diagram (**A1**—**A4**), relative abundance (**B1**,**B2**) and PCoA score plots (**C1**,**C2**) on bac-terial species and fungal genus community in K1 and K2. The odd number represent bacterial species; The even represent fungal species; (**A1**,**A2**) represent K1; (**A3**,**A4**) represent K2.

**Figure 3 foods-11-00012-f003:**
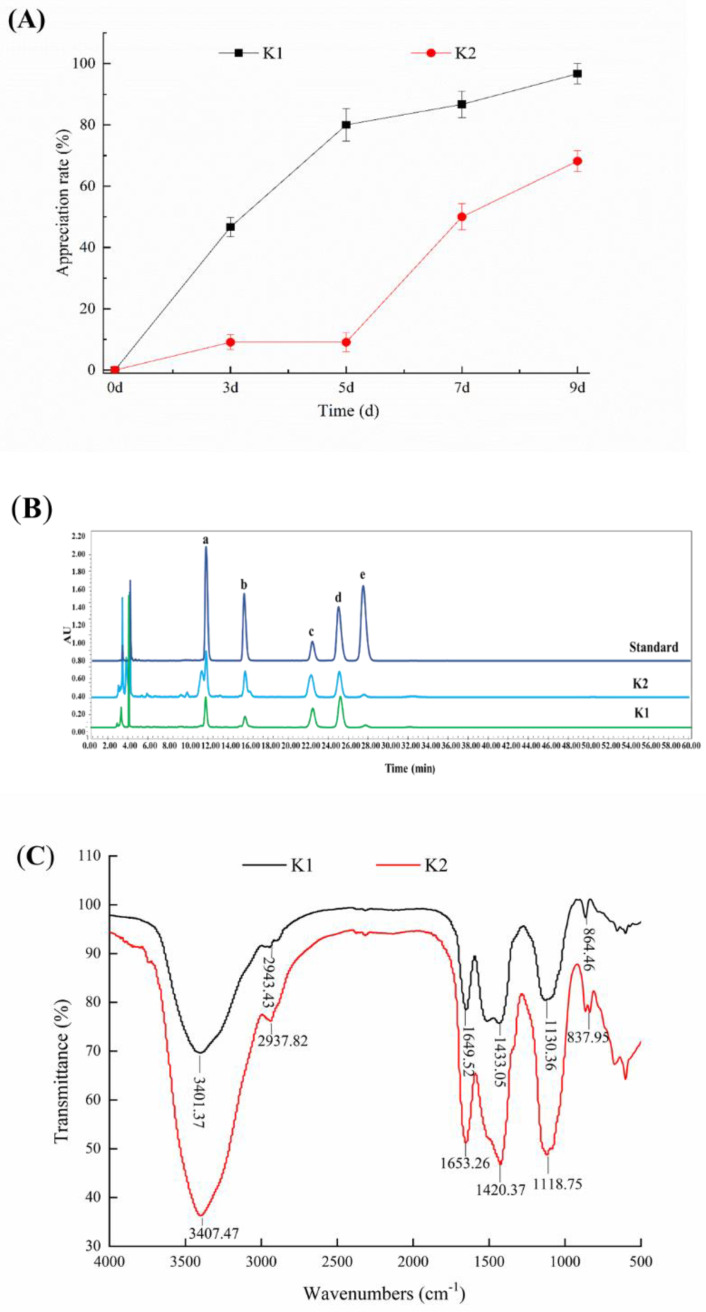
Proliferation of kefir grains (**A**), monosaccharide composition (**B**) and FTIR spectra (**C**) of kefir r-EPS. Monosaccharide standards: mannose (a), rhamnose (b), glucose (c), galactose (d) and arabinose (e).

**Figure 4 foods-11-00012-f004:**
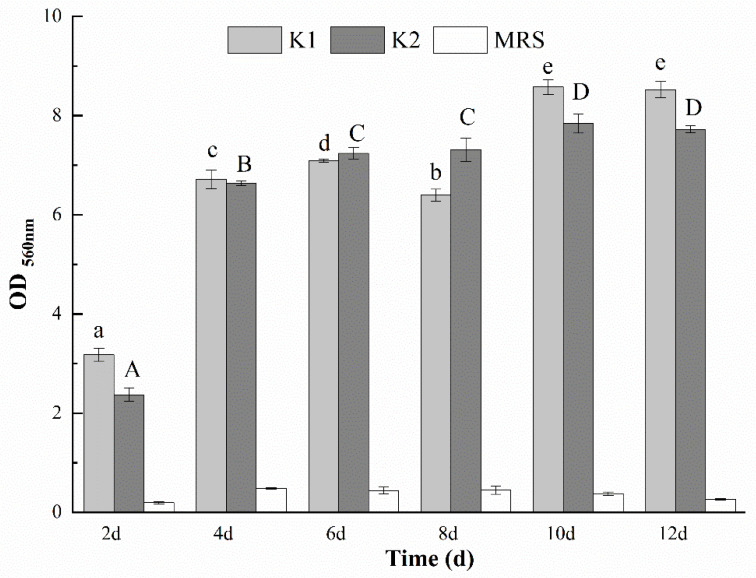
Biofilm-forming ability of kefir grains in the medium with different days. Values with different letters within each cultivation time and OD_560_ nm value of K1 (a–e) and K2 (A–D) are significantly different (*p* < 0.05).

**Figure 5 foods-11-00012-f005:**
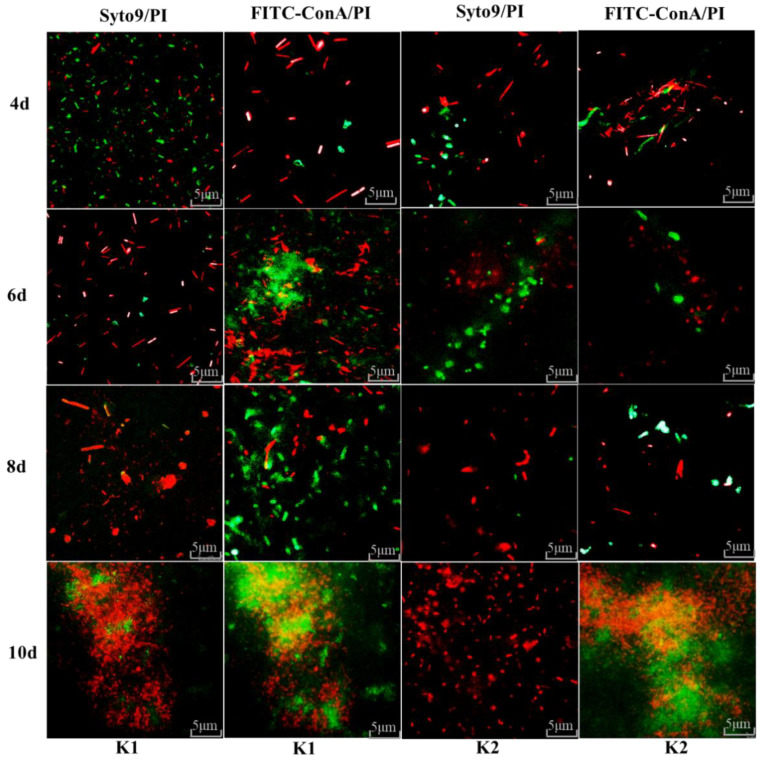
Fluorescence microscope observation on the formation of kefir biofilm. Scale bar = 5 μm. Live cells and dead cells of biofilms were stained with Syto9 (green) and PI (red); the EPS and dead cells of biofilms were stained with FITC-ConA (green) and PI (red).

**Table 1 foods-11-00012-t001:** Sequencing results for the two Tibetan kefir grains and two related kefir milks.

Sample ^a^	Observed OTUs		Number of Sequences		Shannon Index		Chao Index		Simpson Index	
	Bacteria	Fungi	Bacteria	Fungi	Bacteria	Fungi	Bacteria	Fungi	Bacteria	Fungi
K1-G	3	20	37,390	50,045	0.22609	0.1133	23	20	0.45443	0.96659
K1-M	23	6	40,148	74,209	0.31343	0.1891	5	6	0.33823	0.91517
K2-G	4	14	47,988	30,691	1.18108	0.3147	4	21	0.89401	0.84856
K2-M	4	23	43,221	32,009	0.90832	1.3493	4	27	0.83235	0.36902

^a^ K1-G: Linzhi kefir grains; K1-M: Linzhi kefir milks; K2-G: Naqu kefir grains; K2-M: Naqu kefir milks. Shannon index is used to estimate the microbial diversity in the sample, and the value is positively correlated with community diversity; Chao index is used to estimate the number of OTUs contained in the sample, and it is positively correlated with the number of sample species. Simpson index is used to estimate the microbial diversity in the sample, and it is negatively correlated with community diversity.

**Table 2 foods-11-00012-t002:** Strains isolated from K1 and K2.

**Strains (Identification %)**	**Colony Characteristics**	**Separation Sources**	**Medium**
*L. kefiri* K1-M1 (99.93)	Round, dry, transparent, entire	K1-M	MRS
*L. kefiri* K1-M2 (99.97)	Round, moist, white, smooth	K1-M	MRS
*L. kefiranofaciens* K1 (99.96)	Round, moist, white, smooth	K1-M	MRS
*L. kefiranofaciens* G-M1 (99.98)	Round, moist, transparent, smooth	K1-G	MRS
*L. kefiri* G-M2 (99.94)	Round, dry, transparent, entire	K1-G	MRS
*L. kefiranofaciens* subsp *kefiri granum* G-M6 (99.93)	Round, moist, transparent, smooth	K1-G	MRS
*K. marxianus* Y1 (99.95)	Round, ropy, white, smooth	K1-M	PDA
*K. marxianus* G-Y3 (99.93)	Round, ropy, white, entire	K1-G	PDA
**Strains (Identification %)**	**Colony Characteristics**	**Separation Sources**	**Medium**
*E. durans* K2-M3 (99.97)	Round, dry, transparent, entire	K2-M	MRS
*F. fructosus* K2-Y6 (99.94)	Punctirorm, ropy, white, smooth	K2-M	MRS
*L. kefiri* K2-MY1 (99.94)	Round, moist, white, smooth	K2-M	MRS
*L. paracasei* K2-MX1 (99.93)	Round, moist, transparent, smooth	K2-M	MRS
*K. marxianus* K2-Y1 (99.98)	Round, ropy, white, entire	K2-M	PDA
*A. fabarum* K2-Y4 (99.96)	Punctirorm, ropy, transparent, smooth	K2-M	MRS
*L. kefiri* K2-GM5 (99.94)	Round, dry, transparent, entire	K2-G	MRS
*E. durans* K2-M2 (99.96)	Punctirorm, ropy, white, smooth	K2-G	MRS
*K. marxianus* K2-Y3 (99.97)	Round, ropy, white, entire	K2-G	PDA

## Data Availability

Not applicable.

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
