# Peer review of "The Microbial Diversity and Biofilm-Forming Characteristic of Two Traditional Tibetan Kefir Grains"

_foods, 2021, doi:10.3390/foods11010012_

Round 1
Reviewer 1 Report
- Line 42-44 it is necessary to include a reference.
- Figure 1. I advice changing or improving the nomenclature and footnote to clarify the figure. Include scale bars in all images.
- Line 253: change K2 for K2-M.
- Line 264 and 265: K. marxianus and K. lactis in italic letters.
- Line 268: Instead of K. unispora, I think text refers to K. turicensis.
- Figure 3: The footnote is incorrect, image B corresponds to the monosaccharide composition and image C whit FTIR spectra.
Author Response
Reviewer #1
Response: Many thanks for the reviewer’s concern and detailed suggestions on our manuscript. We have revised our manuscript according to the suggestions of the reviewer. Revised portions are marked in red in the manuscript. Detailed reply to the comments and suggestions has been made as follows:
1 Line 42-44 it is necessary to include a reference.
Response: We thank the constructive comments on our manuscript. We have included a referenced in Line 42-44.
2 Figure 1. I advise changing or improving the nomenclature and footnote to clarify the figure. Include scale bars in all images.
Response: Thanks for your useful comment. We have improved the footnote and added the scale bars in Figure.1 in Line 209-210, and we hope it could make you satisfy.
3 Line 253: change K2 for K2-M.
Response: Thanks for your constructive comment. We have changed K2 for K2-M in line 254.
4 Line 264 and 265: K. marxianus and K. lactis in italic letters.
Response: Thank you very much for your comment. We have modified K. marxianus and K. lactis to italic letters in line 266.
5 Line 268: Instead of K. unispora, I think text refers to K. turicensis.
Response: Thank you for your helpful comments. We have changed “K. unispora” to “K. turicensis” in Line 270.
6 Figure 3: The footnote is incorrect, and image B corresponds to the monosaccharide composition and image C whit FTIR spectra.
Response: Thanks for your valuable suggestion. We have modified the footnote of “Proliferation of kefir grains (A), monosaccharide composition (B) and FTIR spectra (C) of kefir r-EPS. Monosaccharide standards: mannose (a), rhamnose (b), glucose (c), galactose (d) and arabinose (e).”
Reviewer 2 Report
The relevance and interest of the study is clear and well demonstrated.
Although the study is correctly founded, both with regard to the introduction and the discussion of the results, it is nevertheless suggested to update the bibliography used, given that 60% of the references presented are more than 5 years old.
Author Response
Reviewer #2
The relevance and interest of the study is clear and well demonstrated. Although the study is correctly founded, both with regard to the introduction and the discussion of the results, it is nevertheless suggested to update the bliography used, given that 60% of the references presented are more than 5 years old.
Response: We highly appreciate the editors' and reviewers' kind consideration of the scientific content of our work. We have studied comments carefully and have made correction which we hope to meet with approval. We have updated the references marked in red as much as possible to make the manuscript more complete.
Reviewer 3 Report
A well-written article with a detailed methodology that makes it possible to repeat experiments of the research. Below I’m presenting suggestions which, if introduced, will improve the quality of the manuscript.
- „An excellent composition of …” -> what do the authors mean by this? (it is not clear) [line 37]
- “yeasts (Kluyveromyces, Candida and Saccharomyces)” -> proposal: yeasts (Kluyveromyces, Saccharomyces and nonpathogenic strains of Candida) [line 38]
- “Candidahumilis”-> Candida humilis [line 44]
- “as section of 2.1 mentioned” -> as mentioned in the 2.1 section [line 115]
- please make a space before sections 2.6.1 and 2.6.2
- “used to dye the membrane” -> used to dye bacterial cells [line 185] (to be more specific these pigments react with bacterial genetic material not membranes)
- Below Table 1, it is worth adding a legend with a description and limit values of the Shannon, Chao and Simpson indexes. Many readers will not be able to interpret this data on their own.
- “In addition to the large and variable …” -> it is worth adding a new paragraph before this sentence (this is a new thought, and the whole paragraph without it is too long) [line 256]
- Graphs A1, A2, A3, A4, B1 and B2 in the "Figure 2" panel should be significantly enlarged (One graph in a row). After printing, you cannot read anything.
- “determined after 10 days of cultivation” -> determined after 4, 6, 8 and 10 days of cultivation [line 383]
- “EPS on biofilm” -> EPS in biofilm [line 402]
- “the formation of biofilms was ultimately affected the proportion of” -> Please remove the empty space [line 416]
- “Figure 5. Fluorescence microscope observation on the formation of kefir biofilm. Scale bar = 5 μm.” -> Requires a legend with a description of the fluorescent dyes and the components they stain
- Conclusions: Is it possible to choose which bacteria influence these parameters? It could be helpful in creating new, important formulations with desirable characteristics.
Author Response
Reviewer #3
A well-written article with a detailed methodology that makes it possible to repeat experiments of the research. Below I'm presenting suggestions which, if introduced, will improve the quality of the manuscript.
Response: We gratefully thank the editor and all reviewers for taking their time to make constructive remarks and useful suggestions on our manuscript. Those comments are all valuable and very helpful for revising and improving our paper. We have read the reviewer’s comments carefully and have made revisions which are marked in red in the revised manuscript.
1 “An excellent composition of" > what do the authors mean by this? (it is not clear) [line 37] "yeasts (Kluyveromyces, Candida and Saccharomyces)" > proposal: yeasts (Kluyveromyces, Saccharomyces and nonpathogenic strains of Candida) [line 38]
Response: We are very grateful to the editor for the good advice. We have modified line 37-38 to: A composition of yeasts (Kluyveromyces, Saccharomyces and nonpathogenic strains of Candida) and some AAB were also revealed in different kefir grains.
2 "Candidahumilis"-> Candida humilis [line 44]
Response: Thanks for your constructive comment. We have changed “Candidahumilis” to “Candida humilis” in line 43.
3 "as section of 2.1 mentioned" > as mentioned in the 2.1 section [line 115]
Response: Many thanks for the reviewer’s concern and detailed suggestions on our manuscript. We have changed "as section of 2.1 mentioned" to “as mentioned in the 2.1 section” in line 113.
4 please make a space before sections 2.6.1 and 2.6.2.
Response: Thank you for your suggestion. We have refined the space before sections 2.6.1 and 2.6.2.
5 "used to dye the membrane" > used to dye bacterial cells [line 185] (to be more specific these pigments react with bacterial genetic material not membranes)
Response: Thanks for reviewer’s detailed explanation, we have corrected it to “used to dye bacterial cells” in line 183.
6 Below Table 1, it is worth adding a legend with a description and limit values of the Shannon, Chao and Simpson indexes. Many readers will not be able to interpret this data on their own.
Response: Thank you for your suggestion. We have added the legend of Shannon, Chao and Simpson indexes in Line 226-229.
7 "In addition to the large and variable."> it is worth adding a new paragraph before this sentence (this is a new thought, and the whole paragraph without it is too long) [line 256]
Response: Thank you for your suggestion. We have added a new paragraph before the sentence in line 258.
8 Graphs A1, A2, A3, A4, B1 and B2 in the "Figure 2" panel should be significantly enlarged (One graph in a row). After printing, you cannot read anything.
Response: Thanks for your useful comment. We have enlarged graphs A1, A2, A3, A4, B1 and B2 in a row to read more clearly.
9 "determined after 10 days of cultivation" > determined after 4, 6, 8 and 10 days of cultivation [line 383]
Response: Thanks for your constructive comment. We have corrected "determined after 10 days of cultivation" to “determined after 4, 6, 8 and 10 days of cultivation”.
10 “EPS on biofilm" > EPS in biofilm [line 402]
Response: Thanks for your valuable suggestion. We have refined the “EPS on biofilm" to “EPS in biofilm”.
11 "the formation of bioflms was ultimately affected the proportion of" > Please remove the empty space [line 416]
Response: Thank you for your helpful comments. We have removed the empty space.
12 “Figure 5. Fluorescence microscope observation on the formation of kefr biofilm. Scale bar= 5 um." > Requires a legend with a description of the fluorescent dyes and the components they stain.
Response: Thanks for your constructive comment. We added a legend of “Live cells and dead cells of biofilms were stained with Syto9 (green) and PI (red); the EPS and dead cells of biofilms were stained with FITC-ConA (green) and PI (red).” in line 421-423.
13 Conclusions: Is it possible to choose which bacteria influencing these parameters? It could be helpful in creating new, important formulations with desirable characteristics.
Response: We thank the constructive comments on our manuscript. We really need to pay attention to the ratios of bacteria in the future to create a new important desirable characteristics of kefir grains. So we have added a prospect about the suggestion in the conclusion.